# Effects of the Action for Neutralization of Bullying Program on Bullying in Spanish Schoolchildren

**DOI:** 10.3390/ijerph18136898

**Published:** 2021-06-27

**Authors:** Ana Martínez-Martínez, David Pineda, Manuel Galán, Juan C. Marzo, José A. Piqueras

**Affiliations:** Department of Health Psychology, Faculty of Social and Health Sciences, Forensic Psychology Unit, Center for Applied Psychology, Campus of Elche, Miguel Hernandez University (UMH), 03202 Elche, Spain; ana.martinezm@umh.es (A.M.-M.); galanmorillomanuel@gmail.com (M.G.); jc.marzo@umh.es (J.C.M.); jpiqueras@umh.es (J.A.P.)

**Keywords:** bullying, schoolchildren, bullying program, effectiveness evaluation

## Abstract

Bullying can have serious physical and emotional consequences. In recent years, interest in this phenomenon has been growing, becoming a public health problem in the first world. The aim of this study was to evaluate the effects of the Action for Neutralization of Bullying Program (ANA) in Spanish children. This study used a quasi-experimental design that included a pre-test evaluation, 2 months of intervention, a post-test, and 3 months of follow-up. A sample of 330 children aged 7–12 years (M = 9.27; SD = 1.09) from third to sixth grade participated in the study. One hundred and fifty-nine were girls (48.2%). The program consisted of eight group sessions in which empathy, assertiveness, communication skills, conflict resolution, and group cohesion were worked on. The results showed statistically significant reductions in verbal abuse behaviors (*t* = 4.76, *p* < 0.001), direct social exclusion (*t* = 3.53, *p* < 0.001), threats (*t* = 2.04, *p* = 0.042), aggression with objects (*t* = 3.21, *p* < 0.001), and physical abuse (*t* = 4.41, *p* < 0.001). The differences were not statistically significant for indirect social exclusion behaviors (*t* = 1.86, *p* = 0.065) or cyberbullying (*t* = 0.31, *p* = 0.756). The effects in the reduction of the bullying behaviors decreased after the implementation of the program, achieving even greater reduction in victimization behaviors after 3 months than immediately after the end of the program. These results indicate that the ANA program is effective in reducing bullying behaviors in a group of children. Implications for practice and future research are discussed.

## 1. Introduction

Bullying has been defined as a distinct type of aggression characterized by repeated and systematic abuse of power [1,2]. This abuse can include physical, verbal, and relational violence, and cyberbullying [3]. In the dynamics of bullying, at least three roles have been identified: bully (i.e., a person who actively perpetrates the physical and/or psychological harassment against the victim), victim (i.e., a person who is physically or/and psychologically harmed and who suffers by being continuedly intimidated), and observer or bystander (i.e., a person who passively witnesses and who can have some influence in the bullying behaviors). These roles are not always distinct from one another, as they tend to be more of a continuum rather than three separate categories [1,4,5,6]. However, bullying is an extremely harmful aggressive behavior that can begin in early childhood and continue over time throughout the school years [7]. Research suggests that between 10% and 30% of children and youth are involved in bullying, although prevalence rates vary significantly depending on how the bullying is measured and by gender [7,8,9].

### 1.1. Effects of Bullying on Well-Being and Health

Acknowledging and knowing the effects of bullying is vital to understand the necessity to implement programs to prevent it from happening. Bullying is a detrimental factor against mental health and well-being at physical, mental, and social-emotional levels in children [10,11]. Children who are bullied can be affected by it in later stages of life, such as adolescence and adulthood [10,12]. The social-emotional consequences suffered by victims of bullying can be divided into internalizing and externalizing problems [11,13]. Among the internalizing problems suffered by victims of bullying, depressive symptoms, anxiety, and feelings of loneliness can lead to suicidal thoughts and behaviors [13,14,15]. Externalizing problems include aggressive behavior or conduct disorders, substance abuse, or self-injurious behaviors, among others [11,16]. It is very important to note that some of these negative consequences of bullying not only arise in those who are victims of bullying, but also in those who commit it, with even higher prevalence rates observed in minors who both bully and are bullied by others [16].

### 1.2. Anticipating Bullying

In recent years, there has been growing awareness of the need to anticipate the occurrence of this type of school violence. Many studies have focused on identifying the important contextual and individual factors [17]. Some important contextual factors are family environment, school climate, community, peer status, and peer influence, and some important individual factors are internalized and externalized behavior, cognitions, and academic performance. Special attention must be given to the contextual factors, and it has been proposed that the most powerful factors for bullying are parental and peer influences, as well as community factors. This contextual aspect relies on the fact that previous research has shown how antisocial behaviors (including bullying) are enhanced by a context with pro-criminal attitudes that has developed some degree of tolerance to these delinquent behaviors. Thus, due to this normalization, this lifestyle remains [6,17,18]. 

On the other hand, peer status has been found to have the greatest effect size as a risk factor for bullying victimization. In terms of the individual factors, for bullies, it has been found that externalized problems, related cognitions, and low academic performance are the most powerful risk factors for bullying perpetration. However, for victims, internalized problems and lack of social competence appear to be the strongest risk factors [17]. Socioemotional competencies are defined as the set of knowledge, skills, and attitudes necessary to understand, express, and appropriately regulate emotional phenomena [19]. Deficiencies in these social-emotional competencies are closely related to aggressive bullying behaviors [10,20].

### 1.3. Effectiveness of Anti-Bullying Programs in Schools

Much of the meta-analytical work has not been particularly optimistic about the results offered by previous intervention programs, concluding that the effect sizes of the interventions are generally small or very small [21,22]. Some of these effect sizes fall short of clinical significance (*r* = 0.12) for self-reported victimization and bullying measures [23]. Furthermore, this measure may be influenced by publication bias, that is, the tendency of journals and researchers to publish only statistically significant results, so the size of the expected effect for these programs may be even smaller [24,25]. The main problems encountered in the effectiveness of these programs have been related to their implementation. Specifically, students have doubts about the credibility of a program when those who present it are external to the context of the situation. Another relevant factor is the fact that the people in charge of the program’s implementation will be in the center for a limited time, which means that many episodes of bullying will not be detected or will not have a response, or the response will not be fast enough [26]. This greatly affects student engagement in the project and, therefore, its effectiveness.

### 1.4. The Action for Neutralization of Bullying Program

The Action for Neutralization of Bullying program (ANA, for its Spanish acronym: *Acción para la Neutralización del Acoso*) applies components that have been shown to be effective in previous research; specifically, it promotes the development of empathy [27], assertiveness [28], communication skills [29], conflict resolution [30], group cohesion [31], and values in favor of coexistence and nonviolence [32]. This intervention was developed by the authors and is based on previous theoretical and research work. As a whole, the program’s components focus on changing attitudes toward violence and promoting the development of values that support coexistence. The program tries to solve the implementation problems reported by previous studies, giving special weight to observers, with special emphasis on observers who maintain a certain degree of authority in the school and who interact with the children, such as the educational community (teachers, administration, and service workers). This special weight given to the observers relies on the fact that they are the ones constantly in contact with the children and consequently the ones that might be able not only to apply the program effectively over the years or the different generations, but also to maintain it. Moreover, this role given to the observers offers the possibility to, albeit following the program, adapt it to specific necessities of the children [33]. The intervention applied in this study consisted of eight group sessions delivered over two months and was aimed mainly at mobilizing the observers. Three training sessions were held for the educational community and two were held for the parents. Both trainings were conducted before the start of the children’s training program. In both cases, the objective was to present the training program to be carried out with the children, to explain the measures that would be taken and to ask for their collaboration in condemning violence in any of its manifestations. Children were not allowed to attend these sessions. The sessions were organized as follows:

Session 1. Theme: Psychoeducation. Activities: Introduction to the program, guidelines and norms, definition of bullying, agents involved in bullying, the dynamics of bullying, recognizing bullying situations (group dynamics), and thought and closure.

Session 2. Theme: Empathy. Activities: What empathy is and what it is for, how would I feel if..., identifying emotions (group dynamics), and thought and closure.

Session 3. Theme: Empathy. Activities: summary of the previous session, identifying situations (group dynamics), I congratulate my classmates (group dynamics), and thought and closure.

Session 4. Theme: Assertiveness. Activities: what is assertiveness? What is assertiveness good for? What happens if I am not assertive? I am assertive, I control what happens to me, I congratulate my classmates (group dynamics), and thought and closure.

Session 5. Theme: Communication skills. Activities: What are communication styles? Practicing assertive communication (role paying): eye contact, volume and tone of voice, verbal fluency, posture, gestures, and verbal content of the message. I congratulate my classmates (group dynamics) and thought and closure.

Session 6. Theme: Conflict resolution. Activities: What is a problem? Solving problems, the 5-step method (clarify the problem, look for solutions, evaluate each solution, choose the best solution, and implement the chosen solution), practice classroom situations 1 (role playing), I congratulate my classmates (group dynamics), and thought and closure.

Session 7. Theme: Conflict resolution. Activities: Remembering the 5-step troubleshooting, “Brave people wanted!” (group dynamics), practice classroom situations 2, 3, 4, and 5 (role playing), I congratulate my classmates (group dynamics), to prevent bullying, we expect you to “be brave”, and thought and closure.

Session 8. Theme: Group cohesion. Activities: The ball of wool (group dynamics), revision of topics worked on throughout the program, and thought and closure.

### 1.5. Objectives and Hypotheses

This study aimed to evaluate the effects of the ANA program in a sample of Spanish children. Its hypotheses were as follows: (I) the implementation of the program would reduce the bullying behaviors perceived by the victims, (II) the active involvement of teachers in the implementation of the program would help to maintain the results in the long term, (III) the implementation of the program would improve the social-emotional competencies of the participants, and finally, (IV) the implementation of the program would improve the general welfare of the participants.

## 2. Materials and Methods

### 2.1. Design

This study used a quasi-experimental design and included a pre-test evaluation, 2 months of intervention, a post-test, and 3 months of follow-up. The sample was selected by convenience. Given that the application of the program in its entirety, including follow-up, covered a complete academic year, it was not possible to establish a control group (waiting list) because it would mean that this group would not eventually receive the treatment.

### 2.2. Ethical Concerns

This study was approved by the Bioethics Commission of the University Miguel Hernandez of Elche. The program was presented to the educational center and obtained approval from the center’s administrators. The center obtained informed consent from the parents. At the end of the study, the results were presented to the center’s administrators and to the teachers, children, and parents.

### 2.3. Participants and Procedure

A total of 330 children aged 7–12 years (*M* = 9.27; *SD* = 1.09) from third to sixth grade participated in the study: 159 were girls (48.2%) and 171 were boys (51.8%). Participants were recruited from a school located at Elche. Socioeconomic status of the school students (obtained at a different time point) is rated between intermediate and high according to the Family Affluence Scale (FAS [34]) and between medium and mid–high in the Hollingshead Four-Factor Index of Socioeconomic Status (SES [35]) Data were collected at three time points: prior to the application of the ANA program, at the end of the program, and at the 3-month follow-up. The program was carried out for 2 months.

### 2.4. Instruments

To assess the program’s effectiveness in reducing bullying behavior, the self-reported Peer Bullying Questionnaire was used [36]. Although the Peer Bullying Questionnaire includes several scales that assess different aspects of bullying among peers, this study only used the scale related to bullying behavior. That scale consists of 39 items on the various forms of bullying that the child or youth might have experienced from their peers. The scale separately evaluates the following forms of peer bullying: verbal abuse (11 items, e.g., “I am insulted by other children”), direct social exclusion (5 items, e.g., “They tell others not to be or not to talk with me”), threats (4 items, e.g., “They threaten to beat me”), cyberbullying (4 items, e.g., “When I chat with other children, they mess with me”), indirect social exclusion (4 items, e.g., “They forbid others to play with me”), object-based aggression (3 items, e.g., “They throw things at me (class objects, paper balls, rocks, etc.)”), and physical abuse (8 items, e.g., “They pull my hair”). Answers are based on a three-point frequency scale that ranges from 0 (“Never”) to 2 (“Many times”). The scale has demonstrated adequate psychometric properties in past research [36]. In the present sample, the following Cronbach’s alpha coefficients were obtained: 0.84 for verbal abuse, 0.72 for direct social exclusion, 0.57 for threats, 0.62 for cyberbullying, 0.61 for indirect social exclusion, 0.50 for object-based aggression, and 0.79 for physical abuse, showing values very similar to those previously reported by the authors [36].

To assess the extent to which the application of the program can improve participants’ social-emotional competencies, the Spanish version of the Social Emotional Health Survey-Primary-Revised was applied [37,38]. The scale is a self-reported, 29-item instrument with 6-point Likert scale responses ranging from 1 (“no”) to 6 (“always”). The scale measures five areas related to youth well-being and school participation. The five subscales are gratitude, zest, optimism, persistence, and prosocial behavior. Some representative items from the subscales are: “I am lucky to go to my school” for gratitude, “I expect good things to happen at my school” for optimism, “I get excited when I learn something new at school” for zest, and “I keep working until I get my schoolwork right” for persistence. The total score from the first four subscales provides a score of co-vitality (a multidimensional, higher-order construct, which includes a range of social and emotional psychological dispositions that are hypothesized to be associated with positive youth development) for students. Previous studies have reported reliability coefficients of the Spanish version ranging from α = 0.73 to α = 0.84 for the five factors and α = 0.91 for the co-vitality factor. The reliability coefficients obtained in this study were α = 0.82 for gratitude, α = 0.70 for optimism, α = 0.83 for zest, α = 0.70 for persistence, α = 0.84 for prosocial behavior, and α = 0.88 for the co-vitality factor.

To assess the program’s effect on the overall well-being of the children, the KIDSCREEN-10 index [39] was applied. The KIDSCREEN-10 index is a 10-item questionnaire that assesses the subjective health-related quality of life and well-being of children and adolescents aged 8 to 18 years. Dimensions of the scale include affective symptoms of depressed mood, cognitive symptoms of disturbed concentration, psycho-vegetative aspects of vitality, energy, and feeling well, and psychosocial aspects correlated with mental health, such as the ability to experience fun with friends or getting along well at school. Some examples of KIDSCREEN-10 items are “Thinking about the last week, have you felt full of energy?” or “Thinking about the last week, have you felt lonely?”. In this study, the internal consistency was 0.79.

### 2.5. Data Analysis

Descriptive statistics such as mean (*M*) and standard deviation (*SD*) were calculated. To find differences, Student’s and Fisher’s tests were applied. Following Cohen’s [40] suggestions, we assume that small, medium, and large effects would be reflected in values of partial η^2^ equal to 0.009, 0.059, and 0.138, respectively.

## 3. Results

Table 1 shows the descriptive statistics of the different bullying behaviors evaluated by gender and for the group as a whole. Differences between genders can be seen in the pre-test, with higher scores for boys in all bullying behaviors, except for indirect social exclusion, where girls score slightly higher. However, these differences were only statistically significant for direct social exclusion (*F*_(1, 328)_ = 4.42; *p* = 0.03; η^2^ = 0.014) and physical abuse (*F*_(1, 328)_ = 24.36; *p* < 0.001; η^2^ = 0.074).

Table 2 shows the descriptive statistics of the social-emotional competencies and the levels of subjective well-being of the participants.

### 3.1. Program Effects

To assess the effects of the program (hypothesis I), the differences between the pre-test and post-test were calculated for each of the bullying behaviors. There were statistically significant reductions in verbal abuse behaviors (*t* = 4.76, *p* < 0.001), direct social exclusion (*t* = 3.53, *p* < 0.001), threats (*t* = 2.04, *p* = 0.042), aggression with objects (*t* = 3.21, *p* < 0.001), and physical abuse (*t* = 4.41, *p* < 0.001). The differences were not statistically significant for indirect social exclusion behaviors (*t* = 1.86, *p* = 0.065) or cyberbullying (*t* = 0.31, *p* = 0.756). No significant differences were found for the Spanish version of the Social-Emotional Health Survey-Primary-Revised (*p* < 0.001). The differences were statistically significant for the KIDSCREEN-10 index (*F* = 12.95, *p* < 0.001, η^2^ = 0.05).

### 3.2. Maintaining Results

Figure 1 shows that levels of bullying decreased after the application of the program for all behaviors evaluated and had decreased further by the 3-month follow-up in all behaviors except aggression, which remained at the same level. Figure 2 shows that levels of subjective well-being increased after the program and had increased further by the 3-month follow-up. Finally, Figure 3 shows the changes in social-emotional competencies in the participants across the three time points. Some minimal improvements can be observed in these values.

To assess the “maintenance effect” that teachers might have had on the maintenance of these results, it was determined whether the changes in the observed variability of the mean scores for the three measured time points were statistically significant. For the subscales of the Peer Bullying Questionnaire, the results were as follows: verbal abuse: *F_intra_* _(1, 328)_ = 18.49, *p* < 0.001, η^2^ = 0.12; direct social exclusion: *F_intra_* _(1, 328)_ = 12.28, *p* < 0.001, η^2^ = 0.075; threats: *F_intra_* _(1, 328)_ = 7.37, *p* = 0.001, η^2^ = 0.05; cyberbullying: *F_intra_* _(1, 328)_ = 1.59, *p* = 0.21, η^2^ = 0.01; indirect social exclusion: *F_intra_* _(1, 328)_ = 7.08, *p* < 0.001, η^2^ = 0.05; object-based aggression: *F_intra_* _(1, 328)_ = 4.79, *p* = 0.009, η^2^ = 0.03; physical abuse: *F_intra_* _(1, 328)_ = 18.87, *p* < 0.001, η^2^ = 0.11. For the subjective health and well-being variable measured with the KIDSCREEN-10 index, the results were *F_intra_* _(1, 328)_ = 12.29, *p* < 0.001, η^2^ = 0.10. Finally, for the variables of the Spanish version of the Social-Emotional Health Survey-Primary-Revised, the results were as follows: optimism: *F_intra_* _(1, 328)_ = 7.16, *p* = 0.001, η^2^ = 0.05; gratitude: *F_intra_* _(1, 328)_ = 1.15, *p* = 0.310, η^2^ = 0.01; zest: *F_intra_* _(1, 328)_ = 3.98, *p* = 0.020, η^2^ = 0.03; persistence: *F_intra_* _(1, 328)_ = 1.15, *p* = 0.320, η^2^ = 0.010; prosociality: *F_intra_* _(1, 328)_ = 2.20, *p* = 0.113, η^2^ = 0.02; co-vitality: *F_intra_* _(1, 328)_ = 4.69, *p* = 0.010, η^2^ = 0.04.

## 4. Discussion

The present study evaluated the effectiveness of the ANA program in children between 7 and 12 years of age, ranging from third to sixth grade. After the implementation of the program, a significant reduction in bullying victimization was observed in the participants.

Observable improvements in other programs generally have not been sustained over time [22]; therefore, one of the main objectives of the ANA program is long-term maintenance of the program’s effects on bullying behavior and related victimization. In this study, the ANA program achieved this “maintenance effect” 3 months after implementation, achieving even greater reduction in victimization behaviors after 3 months than immediately after the program. One possible explanation for this “maintenance effect” may be the active involvement of teachers and other observers in the educational community during and after the application of the program [33], given the special importance of the program to them. Among the reductions, the decreases after 3 months of reported values of physical and verbal abuse are especially relevant, since these two components of the bullying tend to be the ones with the most weight or more general in these kinds of behaviors [36].

With respect of the direct changes produced after the application of the ANA program, the results show the main reductions in verbal abuse behaviors, threats, aggression with objects, physical abuse, and direct social exclusion behaviors. However, no differences were found in indirect social exclusion and in cyberbullying. This contrast in the changes observed between the first bullying practices mentioned and indirect social exclusion might be explained by the nature of these behaviors. Since the first ones make reference to more observable or direct components of bullying, the second one indicates an indirect component which could be less conscious for the children and thus, more difficult to change [36]. The lack of changes in the cyberbullying behaviors are possibly explained by the low rates of this type of victimization before the implementation of the program, making it very difficult to draw any conclusions from them.

The differences observed in the different types of bullying behavior, based on gender, are consistent with previous literature that also found that boys generally have higher bullying victimization scores, which would imply bigger reductions in these scores [21].

In addition to the reductions observed in victimization, there was also clear improvement in the subjective well-being of the children, which could have been a direct or indirect effect of the decrease in victimization. This effect may be influenced by a decrease in the levels of stress or emotional distress typically reported by children involved in bullying behaviors, both as victims and as aggressors [12,13,16]. Previous studies have found that the long-term effect of this reduction of bullying coupled with improved subjective levels of well-being is related to better adaptation and may help in avoiding or reducing anxiety or depressive symptoms in both the current stage of development and later stages of adolescence or adulthood [16].

Finally, it is worth noting that there were no significant changes in social-emotional competencies. Training in social-emotional competencies, although desirable, is not the main objective of the program. The results suggest that for the improvement of well-being, it is not necessary to provide victims with more coping skills and that it is enough to stop the bullying. However, a slight improvement in these skills was observed, and was most likely due to the relationship between bullying behaviors and these skills [10,20].

## 5. Conclusions

These results indicate that while implementing the ANA program, the bullying behaviors in children decreased at the same that their subjective well-being was increased. One of the strengths of the program is the demonstrated long-term maintenance of its effects after implementation. However, these interesting results achieved by the ANA program have to be interpreted with caution due to some weaknesses related with the study design, as further discussed in the following section.

### Limitations

This study had several limitations. First, because the sample was composed of 330 students from the same center, the results may not be generalizable to other schools with differences in environmental variables (e.g., schools without the high group cohesion of the center, and the high and consistent participation of the observers throughout the program). Second, considering the quasi-experimental design of the study, the effectiveness of the program requires more research due to the lack of a control group. With this design, albeit it leads to some valuable conclusions, it leaves open many alternative explanations for the changes observed in the sample, therefore not allowing to treat the program as the certain causal explanation for the observed changes. Nevertheless, the gains and positive effects produced by the program must be considered. Future research on the program that includes the group of observers in the data collection would allow comparison of the self-reported and perceived levels of bullying and subjective well-being, at least by the observers. In line with the observers, a more contextual approximation would be optimal. Since the problem of the bullying is partially explained by family context, it would be interesting to also assess the prosocial or antisocial tendencies of the children’s families to evaluate their influence in the program changes [6,18]. Additionally, following current trends [33], the inclusion of booster sessions after the end of the program and after an extended period of time has elapsed would be of relevant interest in order to consolidate the gains previously achieved through the program. In addition, longer follow-up would allow further evaluation of the effects of the program. Moreover, it would be interesting to test the individual influence of each of the different ANA program sessions, pursuing a reduction of its length without losing its effectiveness

Another issue that is a general problem of these types of program evaluation research studies, based solely on survey instruments, is that these instruments have the problem that they lack reliability across contexts [41]. In this sense, we believe that this limitation is partially controlled by the fact that the measure used has been validated in the same context as the present research [36].

## Figures and Tables

**Figure 1 ijerph-18-06898-f001:**
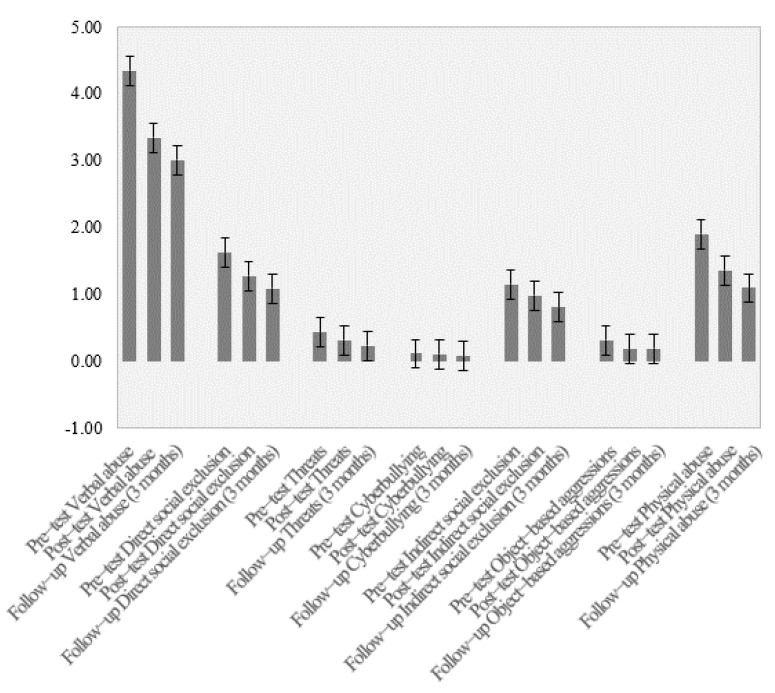
Mean scores in different bullying victimization behaviors for the pre-test, post-test, and 3-month follow-up.

**Figure 2 ijerph-18-06898-f002:**
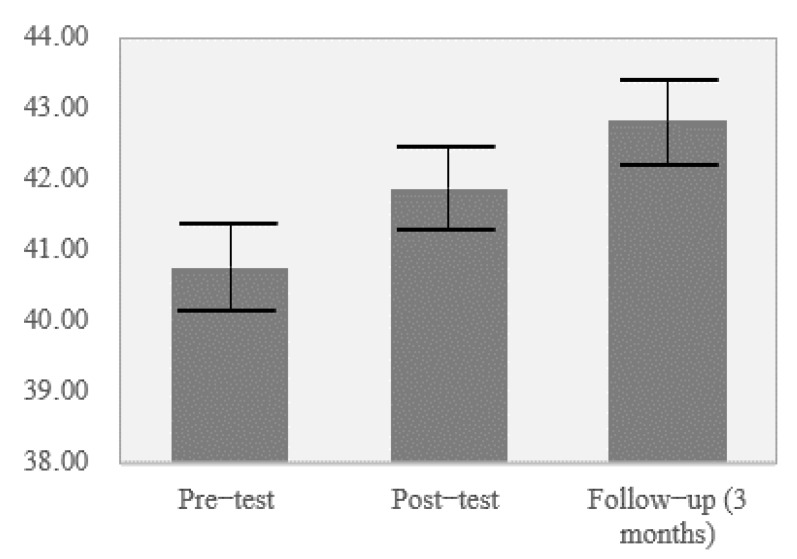
Mean scores in health and subjective well-being for the pre-test, post-test, and 3-month follow-up.

**Figure 3 ijerph-18-06898-f003:**
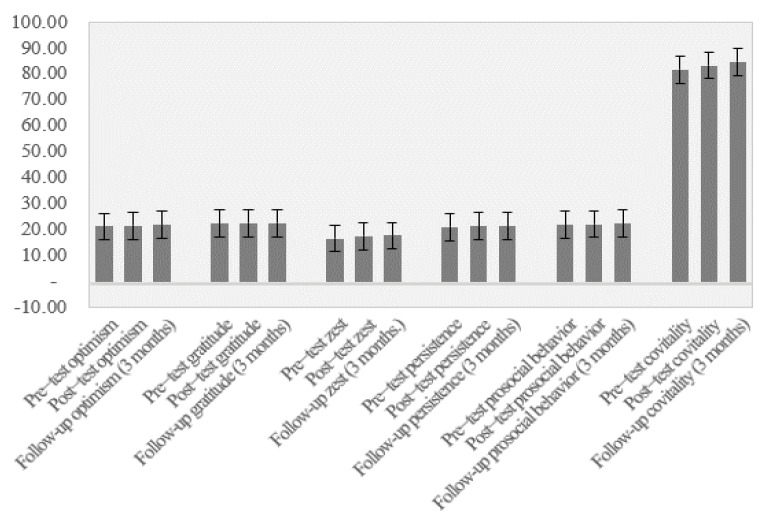
Mean scores for the subscales of the Spanish version of the Social-Emotional Health Survey-Primary-Revised for pre-test, post-test, and the 3-month follow-up.

**Table 1 ijerph-18-06898-t001:** Descriptive statistics of bullying behaviors in the pre-test, post-test, and follow-up ^a^.

Scale	Number of Items	Range	Girls	Boys	Total
			*M (SD)*	*M (SD)*	*M (SD)*
Verbal abuse ^a^	11	[0–22]	3.64 (0.33)	4.44 (0.35)	4.56 (3.90)
Verbal abuse ^b^			2.77 (0.29)	3.51 (0.35)	3.56 (3.72)
Verbal abuse ^c^			2.53 (0.29)	3.18 (0.33)	3.23 (3.80)
Direct social exclusion ^a^	5	[0–10]	1.18 (0.14)	1.81 (0.18)	1.64 (1.85)
Direct social exclusion ^b^			0.82 (0.13)	1.47 (0.17)	1.30 (1.78)
Direct social exclusion ^c^			0.78 (0.13)	1.27 (0.16)	1.17 (1.79)
Threats ^a^	4	[0–8]	0.29 (0.05)	0.42 (0.09)	0.43 (0.90)
Threats ^b^			0.12 (0.03)	0.39 (0.08)	0.33 (0.87)
Threats ^c^			0.06 (0.02)	0.27 (0.07)	0.29 (0.93)
Cyberbullying ^a^	4	[0–8]	0.06 (0.03)	0.14 (0.05)	0.12 (0.49)
Cyberbullying ^b^			0.00 (0.00)	0.12 (0.04)	0.09 (0.50)
Cyberbullying ^c^			0.01 (0.01)	0.05 (0.03)	0.06 (0.38)
Indirect social exclusion ^b^	4	[0–8]	1.01 (0.12)	0.95 (0.12)	1.17 (1.47)
Indirect social exclusion ^c^			0.81 (0.10)	1.00 (0.13)	0.98 (1.35)
Indirect social exclusion ^d^			0.71 (0.09)	0.73 (0.11)	0.84 (1.28)
Object-based aggressions ^b^	3	[0–6]	0.19 (0.04)	0.26 (0.06)	0.33 (0.77)
Object-based aggressions ^c^			0.09 (0.03)	0.20 (0.05)	0.17 (0.49)
Object-based aggressions ^d^			0.14 (0.04)	0.21 (0.06)	0.22 (0.70)
Physical abuse ^a^	8	[0–16]	1.25 (0.16)	2.31 (0.24)	1.97 (2.32)
Physical abuse ^b^			0.73 (0.13)	1.81 (0.21)	1.43 (2.06)
Physical abuse ^c^			0.56 (0.12)	1.35 (0.19)	1.28 (2.24)

Note. *M* = Mean, *SD* = standard deviation. ^a^ Evaluated with the Peer Bullying Questionnaire [36]. ^b^ Pre-treatment measure. ^c^ Post-treatment measure. ^d^ 3-month follow-up.

**Table 2 ijerph-18-06898-t002:** Descriptive statistics of levels of subjective well-being and social-emotional competencies from the pre-test, post-test, and follow-up ^a, b^.

Scale	Range	Girls	Boys	Total
		*M (SD)*	*M (SD)*	*M (SD)*
Optimism ^c^	[6–24]	21.38 (3.02)	20.66 (3.57)	21.13 (3.26)
Optimism ^d^		21.78 (2.88)	20.99 (4.13)	21.46 (3.53)
Optimism ^e^		22.34 (2.61)	20.64 (4.32)	21.76 (3.22)
Gratitude ^c^	[6–24]	22.68 (2.33)	22.66 (3.60)	22.22 (3.03)
Gratitude ^d^		22.77 (2.10)	21.80 (3.60)	22.37 (2.85)
Gratitude ^e^		23.01 (1.80)	21.68 (3.81)	22.51 (2.88)
Zest ^c^	[6–24]	17.07 (5.33)	16.02 (5.14)	16.54 (5.38)
Zest ^d^		17.90 (5.57)	16.78 (5.57)	17.41 (5.59)
Zest ^e^		18.01 (5,80)	17.14 (5.64)	17.65 (5.72)
Persistence ^c^	[6–24]	21.14 (3.40)	20.57 (3.52)	20.98 (3.36)
Persistence ^d^		21.49 (3.18)	20.75 (3.83)	21.28 (3.35)
Persistence ^e^		22.51 (3.04)	20.78 (4.20)	21.33 (3.49)
Prosociality ^c^	[6–24]	22.26 (2.68)	21.17 (3.31)	21.72 (3.02)
Prosociality ^d^		22.71 (2.18)	21.15 (3.60)	22.05 (2.96)
Prosociality ^e^		22.74 (2.09)	21.26 (3.95)	22.22 (3.11)
Co-vitality ^c^	[30–120]	82.65 (11.55)	79.21 (12.33)	81.65 (11.71)
Co-vitality ^d^		84.07 (11.01)	80.74 (14.12)	83.27 (12.06)
Co-vitality ^e^		85.19 (10.65)	80.38 (15.78)	84.61 (11.53)
Subjective well-being ^c^	[5–50]	41.24 (4.31)	40.07 (5.41)	40.76 (4.89)
Subjective well-being ^d^		41.90 (4.53)	41.01 (5.52)	41.86 (4.8)
Subjective well-being ^e^		42.84 (3.76)	42.02 (4.73)	42.82 (4.00)

Note. *M* = Mean, *SD* = standard deviation. ^a^ Subjective well-being evaluated with the Spanish version of the Social-Emotional Health Survey-Primary-Revised [37]. ^b^ Social-emotional competencies evaluated with the KIDSCREEN-10 index [39]. ^c^ Pre-treatment measure. ^d^ Post-treatment measure. ^e^ 3-month follow-up.

## Data Availability

The data presented in this study are available on request from the corresponding author. The data are not publicly available due to participants privacy (pseudonymization of the data).

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
