# Peer review of "Effects of the Action for Neutralization of Bullying Program on Bullying in Spanish Schoolchildren"

_ijerph, 2021, doi:10.3390/ijerph18136898_

Round 1

Reviewer 1 Report

My field isn’t education and I don’t have first hand experience in dealing with bullying but was wanted to read this paper because I am a psychologist who is interested in how children influence one another and how the culture of a population of children forms and changes. I am therefore not knowledgeable in the literature on bullying and how to combat it but I am familiar with some theoretical literature that might be interesting.

First of all, there were three places in the paper where I think the authors meant should consider their word choice. My training in biology made me familiar with the word “autolysis” (on line 53) as something that occurs in cells – the cells destroy themselves. I haven’t, however, heard the world applied to humans or other multicellular organisms. It’s a good word but perhaps other readers may also not be familiar with it used in this way. The use of the word “predict” on line 58 also confuses me. Again, I can see from the context what you mean but the word “predict” doesn’t seem right to me probably because it has so many meanings in English. A more precise word or a phrase would be better, I think. Line 178 has the sentence “I am threaten to be beaten”, which is a sentence in the Peer Bullying Questionnaire that was translated from Spanish. The form of this sentence isn’t grammatically correct in English.  It would be correct to say, “They threaten to beat me”.   

In reporting the results, I think you should point out that the instance of cyberbullying was so low in this population of children that your study wasn’t able to evaluate how the ANA programme might affect this behaviour.

In terms of theory, I think that it would help your arguments to draw on the theories of “cultural evolution” – a body of study which looks at how ideas, beliefs, knowledge, behaviours etc. spread through a population. This theory very much supports the approach of including bystanders in the programme and not just the perpetrators and victims of bullying. Even relatively small populations have “a culture” and if a school develops a tolerance of culture, the behaviour will be maintained or increased even if only a small proportion of the population actually do the bullying or are its victim.

In the end, when you discuss limitations and future work, do you think it would be a good idea to look at the effectiveness of the various components of the ANA programme.  It includes eight sessions. It would be nice to know if the number of sessions and themes covered could be reduced and produce the same or better outcomes.

Author Response

We would like to thank reviewer 1 for his comments, which have been of great help in improving the manuscript. I hope that the changes made have covered the reviewer's recommendations.

My field isn’t education and I don’t have first hand experience in dealing with bullying but was wanted to read this paper because I am a psychologist who is interested in how children influence one another and how the culture of a population of children forms and changes. I am therefore not knowledgeable in the literature on bullying and how to combat it but I am familiar with some theoretical literature that might be interesting.

First of all, there were three places in the paper where I think the authors meant should consider their word choice. My training in biology made me familiar with the word “autolysis” (on line 53) as something that occurs in cells – the cells destroy themselves. I haven’t, however, heard the world applied to humans or other multicellular organisms. It’s a good word but perhaps other readers may also not be familiar with it used in this way. The use of the word “predict” on line 58 also confuses me. Again, I can see from the context what you mean but the word “predict” doesn’t seem right to me probably because it has so many meanings in English. A more precise word or a phrase would be better, I think. Line 178 has the sentence “I am threaten to be beaten”, which is a sentence in the Peer Bullying Questionnaire that was translated from Spanish. The form of this sentence isn’t grammatically correct in English.  It would be correct to say, “They threaten to beat me”.   

Thank you for helping us to improve the understanding of the manuscript, we have applied all the corrections and make the changes.

In reporting the results, I think you should point out that the instance of cyberbullying was so low in this population of children that your study wasn’t able to evaluate how the ANA programme might affect this behaviour.

We agree with you, luckily these children did not suffer much from cyberbullying, thus we have added this point to the manuscript. However, we have included it in the discussion section instead of in the results section.

In terms of theory, I think that it would help your arguments to draw on the theories of “cultural evolution” – a body of study which looks at how ideas, beliefs, knowledge, behaviours etc. spread through a population. This theory very much supports the approach of including bystanders in the programme and not just the perpetrators and victims of bullying. Even relatively small populations have “a culture” and if a school develops a tolerance of culture, the behaviour will be maintained or increased even if only a small proportion of the population actually do the bullying or are its victim.

Thank you very much for your comment, we strongly belief in the importance of the bystanders or observers but we in the “first” manuscript we did not give enough attention to them. Therefore, we have extended the literature revision regarding the contextual issue and added it in the introduction, limitations and further aims due to its importance. However, we have not included the idea of a productive culture but context instead. This decision, although arguable, relies in some classical critics made to this delinquency related cultural approach, claiming just one culture with different societies or contexts inside that are driven by some common values (e.g., Durkheim, 1893; Kornhauser, 1978 y Serrano-Maillo, 2019).

In the end, when you discuss limitations and future work, do you think it would be a good idea to look at the effectiveness of the various components of the ANA programme.  It includes eight sessions. It would be nice to know if the number of sessions and themes covered could be reduced and produce the same or better outcomes.

Definitely, this is a very interesting point for making the program more efficient, we have added it.

Reviewer 2 Report

The article deals with a very pertinent and current theme.

It is well written and has a good logic of presentation, demonstrating adequate language and writing.

Some comments:

  1. introduction 35

In the dynamics of bullying, at least three roles have 38

been identified: aggressor, victim, and bystander. These roles are not always distinct from one 39:

- I consider that the authors should define in a footnote the identified roles. Victim, aggressor, observer, so that it is understood to whom they are referring.

1.1 Effects of bullying on well-being and health

- I consider it necessary to clarify the concept the authors use :" bullying on well-being and health", in order to make clear the underlying philosophy of the study.

P147 - research design:

- I agree with the methodology and the characterization of quasi-experimental study. But since this is an evaluation of the effectiveness of a programme (ANA), in my opinion, the use of scales alone, without using a more qualitative technique, such as a focus group with each of the groups involved in the study, in order to validate some of the results systematised by the scale, may lose some of its scientific value, but the methodology followed can be used. Today, the problem of bullying and the aggressive behaviour of young people at school is largely the result of their family contexts, particularly those of families with domestic violence, and it is necessary to consider this variable in studies of this nature.

  1. 278 - 4 Discussion

- In my opinion the discussion of the results should be more in-depth and should be organized by categories of indicators of the Ana program, in order to make clearer the effectiveness of the program and its levels of effectiveness by each category.

P 313- 5 Conclusions

I consider that the conclusion has to be revised, because it should refer beyond the great result of the study, the weak points of the programme to be improved should also be identified, since it is an evaluation of the ANA programme.

Author Response

We would like to thank reviewer 2 for his comments, which have been of great help in improving the manuscript. We hope that the changes made have covered the reviewer's recommendations.

The article deals with a very pertinent and current theme.

It is well written and has a good logic of presentation, demonstrating adequate language and writing.

Thank you very much for your positive feedback, we really appreciate it.

Some comments:

  1. introduction 35

In the dynamics of bullying, at least three roles have 38

been identified: aggressor, victim, and bystander. These roles are not always distinct from one 39:

- I consider that the authors should define in a footnote the identified roles. Victim, aggressor, observer, so that it is understood to whom they are referring.

Thank you for the comment, this will help to clarify the definitions we have used to define the roles. We have embedded the definitions in the text since they were not very extensive.

1.1 Effects of bullying on well-being and health

- I consider it necessary to clarify the concept the authors use :" bullying on well-being and health", in order to make clear the underlying philosophy of the study.

Since the main objective of the intervention is to avoid these effects from appearing, we have added a sentence stating this to make it clear. Additionally, we have added the word mental before health. Thank you for the suggestion.

P147 - research design:

- I agree with the methodology and the characterization of quasi-experimental study. But since this is an evaluation of the effectiveness of a programme (ANA), in my opinion, the use of scales alone, without using a more qualitative technique, such as a focus group with each of the groups involved in the study, in order to validate some of the results systematised by the scale, may lose some of its scientific value, but the methodology followed can be used. Today, the problem of bullying and the aggressive behaviour of young people at school is largely the result of their family contexts, particularly those of families with domestic violence, and it is necessary to consider this variable in studies of this nature.

Thank you for this comment, we agree with you in the technique used. We have extended the literature revision regarding the contextual (or even cultural for some authors) issue and added it in the introduction, limitations and further aims due to its importance.

  1. 278 - 4 Discussion

- In my opinion the discussion of the results should be more in-depth and should be organized by categories of indicators of the Ana program, in order to make clearer the effectiveness of the program and its levels of effectiveness by each category.

Thank you for the comment. In order not to be redundant, we have added a paragraph discussing the observed differences.

P 313- 5 Conclusions

I consider that the conclusion has to be revised, because it should refer beyond the great result of the study, the weak points of the programme to be improved should also be identified, since it is an evaluation of the ANA programme.

In order to address this point and not be redundant with the weak points, we have added a sentence at the end of the conclusion stating that the study has several weakness that are discussed in the limitations section right after the conclusion.

Reviewer 3 Report

It is a very interesting and topical study, whose results show the long-term effectiveness of a programme to reduce bullying. This is a very important aspect to evaluate its application in schools. -The references used are mostly up to date, only a little more than 10 of them are more than 15 years old, an aspect that could be improved by updating them. -Some years of the references do not appear in bold type.

Author Response

We would like to thank reviewer 3 for his comment. We hope that the changes made have covered the reviewer's recommendations

It is a very interesting and topical study, whose results show the long-term effectiveness of a programme to reduce bullying. This is a very important aspect to evaluate its application in schools. -The references used are mostly up to date, only a little more than 10 of them are more than 15 years old, an aspect that could be improved by updating them. -Some years of the references do not appear in bold type.

We have made the suggested changes, boldfacing some of the years that were not in bold type.

About the references, we have added recent references following some other reviewers comments and additionally, we have actualize some of the oldest. And actualized the bullying prevalence rates.